# Importance of Punctual Monitoring to Evaluate the Health Effects of Airborne Particulate Matter

**DOI:** 10.3390/ijerph191710587

**Published:** 2022-08-25

**Authors:** Alberto Izzotti, Paola Spatera, Zumama Khalid, Alessandra Pulliero

**Affiliations:** 1Department of Experimental Medicine, University of Genoa, 16132 Genoa, Italy; 2IRCCS Ospedale Policlinico San Martino, 16132 Genoa, Italy; 3Department of Health Sciences, University of Genoa, 16132 Genoa, Italy

**Keywords:** particulate matter exposure, PM2.5, ultrafine particles, public health, environmental health

## Abstract

Particulate matter (PM) pollution is one of the major public health problems worldwide, given the high mortality attributable to exposure to PM pollution and the high pathogenicity that is found above all in the respiratory, cardiovascular, and neurological systems. The main sources of PM pollution are the daily use of fuels (wood, coal, organic residues) in appliances without emissions abatement systems, industrial emissions, and vehicular traffic. This review aims to investigate the causes of PM pollution and classify the different types of dust based on their size. The health effects of exposure to PM will also be discussed. Particular attention is paid to the measurement method, which is unsuitable in the risk assessment process, as the evaluation of the average PM compared to the evaluation of PM with punctual monitoring significantly underestimates the health risk induced by the achievement of high PM values, even for limited periods of time.

## 1. Introduction

Particulate matter (PM) pollution is a major public health problem today. WHO estimated that 4.3 million deaths per year worldwide are attributable to exposure to PM. This PM is emitted in daily activities due to the use of fuels such as wood, coal, and organic residues in appliances without emission abatement systems [1]. A further 3.7 million deaths are attributed to outdoor PM pollution: in this case, the phenomenon also affects Western European countries, the United States, and Australia, despite the progress made in these areas of the planet in reducing emissions from industry and vehicular traffic [2]. The European Environment Agency estimated that in Italy in 2018, 52,300 premature deaths were associated with long-term exposure to PM2.5, 10,400 to NO_2_, and 3000 to O_3_ [3]. PM, therefore, represents by far the most important airborne pollutant in terms of damage to public health. Air Quality Directive 2008/50/CE established an annual limit value of 40 μg/m^3^ for the annual arithmetic mean of PM10 concentrations and a daily limit value for PM10 of 50 μg/m^3^ not to be exceeded more than 35 times per year (90.4 percentile). Furthermore, an annual limit for PM2.5 (25 μg/m^3^) was set up for all types of environments, including traffic and industrial hotspots, as well as an obligation to reduce future PM2.5 levels (a variable reduction percentage depending on the initial concentrations). Concerning the chemical composition of PM10 (where 10 is the aerodynamic diameter) Directives 2004/107/CE and 2008/50/CE fixed annual target values for As (6 ng/m^3^), Cd (5 ng/m^3^) and Ni (20 ng/m^3^), and an annual limit value for Pb (500 ng/m^3^) [4].

The explicit definition of the fractions PM10, according to Article 2 point 18 and 19 of the cited Directive 2008/50/EC, clarifies that any PM plum/population is not mono-modal, i.e., the characteristic size distribution is spread across orders. Thus, the “coarse mode” PM10 still contains the amount of small enough particles and should not be misinterpreted as “health-save”.

Like PM10, PM2.5 is partly emitted as such directly from sources in the atmosphere (“primary” PM2.5) and partly formed through chemical reactions between other pollutant species (“secondary” PM2.5) [5], the mass concentration of PM2.5 is represented by particles in the size range from about 0.1 μm to about 1 μm. “Secondary” particulate matter formed in the atmosphere from precursor gases by aggregation of smaller particles or by condensation of gases on particles acting as a clot may account for a large proportion of the observed mass concentration.

Combustion processes of coal, wood, and petroleum derivatives release high amounts of fine particles independently from the implant source (cars, heating systems, forest fires, etc.) [6,7].

The WHO sets a mean reference annual value of 25 μg/m^3^ for PM2.5. However, both the daily limit value for PM10 (45 μg/m^3^) and the daily limit value for PM2.5 (15 μg/m^3^) refer to mean values calculated over the relevant time [8]. The fact that the observed mean is an inferential figure is therefore overlooked in current detection approaches. A lower mean value than the reference limits could therefore result from low values averaged over several point peaks. From the point of view of health risk, exceeding threshold values, even on a point-by-point basis, is the real health risk factor. Therefore, the current PM measurement method is relevant for environmental chemistry but is poorly predictive of the related health risk. 

This situation is even more relevant if we consider that the relationship between short-term mortality increases and population exposure to PM10 and PM2.5 has no threshold effect. In fact, this relationship is linear even below the reference thresholds as well documented by studies conducted even on large populations [9]. Interestingly, the conclusions of these studies lead the authors to state that if the current method of PM assessment is continued, even if the reference thresholds are met in the future, most PM-related mortality will persist [10]. In addition, some authors found that the associations of mortality with PM concentrations were slightly stronger with PM2.5 than with PM10 in most countries and regions, which added to the evidence that PM2.5 accounted for a larger proportion of the effects of PM10 and PM2.5 combined [11]. The stronger effects of PM2.5 may also be supported by the abundant evidence that this particulate fraction contains more small particles that can absorb toxic components from the air and penetrate deep into the lungs [12].

The slow reduction in PM10 and NO_2_ levels that has been observed in Europe over the last decade is the result of a reduction in emissions of primary particulate matter and the main precursors of secondary particulate matter (nitrogen oxides, sulfur oxides, ammonia) [13,14].

However, exceedances of the daily limit value for PM10 continue to occur in many urban areas and, for NO_2_, the annual limit at monitoring stations located near major roads with vehicular traffic.

Airborne PM originating from car and truck traffic emits fine PM persisting as air suspension for a long time and spreading according to wind streams [15,16].

About PM2.5, although the annual average concentrations are in most cases below the legal limit value, there are cases of exceedances, particularly in the Po Valley basin, which is one of the most critical areas in Europe. The province of Pavia in the Po Valley presents stagnant wind conditions for long periods that influence the accumulation of air pollution. The Alps act as a barrier to the winds, further favoring the accumulation of pollutants. In summary, these factors contribute to thermal inversion, which creates an increase in air pollutants concentration in the low strata of the atmosphere [17]. 

In addition, when considering the WHO guideline values for population exposure to PM2.5 (5 μg/m^3^ as an annual average), there are few cities where PM2.5 levels are below the recommended value. Consequently, in the presence of meteorological conditions favorable to the accumulation of particles, in flat areas and valleys, PM2.5 levels are rather spatially homogeneous, even at significant distances from the main sources of primary PM and the precursors of the secondary component [18]. 

The aim of this paper is to provide evidence to the fact that, to concretely achieve the goal of reducing short-term mortality associated with exposure to PM, it is necessary to go beyond the concept of average assessment and evolve toward the concept of point assessment, which is more predictive of the real risk to health.

## 2. Particle Size

To date, it has been recognized that the particles that have the greatest impact on human health effects are those with an aerodynamic diameter of less than 10 μm [19]. These particles can penetrate the respiratory tract starting from the nasal passages to the alveoli, deep inside the lungs, due to their excessive penetrability [20]. Particles between about 5 and 10 μm are most likely deposited in the tracheobronchial tree, while those between 1 and 5 μm are deposited in the respiratory bronchioles and alveoli, where gas exchange takes place [21].

These particles can affect gas exchange within the lungs and can even penetrate the lung [22].

Particles smaller than 1 μm, in general, behave similarly to gas molecules and, therefore, will penetrate to the alveoli and can further translocate into cellular tissue and/or the circulatory system [23]. The dimensions of ultrafine dust are comparable to those of biological molecules, while they are larger than those of atoms and considerably smaller than those of human red blood cells or human alveolar macrophages [24].

With Directive 2008/50/EC of the European Parliament and of the Council of 21 May 2008 on ambient air quality and cleaner air for Europe, which has yet to be transposed into national law, the European Union intended to update and bring together in a single text the five legal instruments relating to air quality by integrating the latest developments in the medical and scientific spheres, as well as the most recent experience acquired by member states on air quality management [4,25]. Although mass concentration is a well-established indicator of toxicity underpinning current standards, some recent studies also relate particle number and size to health effects, highlighting the possibility that nanoparticles (mainly elemental carbon, transition metals, and metal dioxide) may penetrate through the respiratory tract and, past the epithelial tissue of the alveoli, reach other organs such as brain, liver, and kidneys, into the lymphatic and blood systems. They, therefore, have different effects on human health from those found for fine and coarse dust, and measures taken for PM10 and PM2.5 may not be sufficient to safeguard human health in these respects [23]. 

## 3. Health Effects of Exposure to Particulate Matter

Atmospheric particulate matter is one of the main environmental health risk factors. The system most affected by particulate matter is the respiratory system, and the most important factor in studying its effects on health is particle size, which determines its ability to penetrate the respiratory tract [26]. Depending on the size, the particles are classified into three fractions:***The inhalable fraction:*** includes all particles that manage to enter through the nostrils and mouth;***The thoracic fraction***: includes particles that manage to pass through the larynx and enter the lungs during inhalation, reaching the tracheo-bronchial region (including the trachea and the cilia);***The respirable fraction***: this includes particles small enough to reach the alveolar region, including the non-ciliated airways and alveolar sacs.

PM10 and PM2.5 are assimilated to the thoracic and respirable fractions, respectively.

Although PM10 is still the most widely reported measurement and the most relevant indicator for most epidemiological data, the WHO Air Quality Guidelines for particulate matter are now mostly based on studies using PM2.5 as the most appropriate indicator. The PM2.5 values of the guidelines are converted into the corresponding PM10 values by applying a PM2.5/PM10 ratio of 0.5. This ratio is typical for urban areas in a developing country, while it is at the lower end of the range found in urban areas in an industrialized country (0.5–0.8). When local standards are established based on collected data, a specific value for this ratio can be derived to better reflect local conditions [27]. 

Many studies, especially in the USA, have reported that it is the fine fraction (PM2.5) in particular that is associated with adverse health effects; based on these considerations, the EPA (Environmental Protection Agency) has decided to establish standard values for PM2.5 as well. 

The literature is unanimous in attributing fundamental importance to monitoring and exposure estimation methods for a correct assessment of the exposure-response function for the characterization of health risks related to airborne pollutants [28]. 

The assessment of exposure to air pollution using only average urban concentrations may lead to an underestimation of the “burden” of adverse health effects attributable to higher concentrations of pollutants that can be detected near the sources, with reference to roads with high traffic density and the contribution of fine and ultrafine dust [29]. Differences in studies make it difficult to draw general conclusions about how the measures work. Detecting changes in population health and air pollution levels is challenging, and assessing whether the changes that occur are due to a specific measure is complex. Air pollution levels change constantly and often unpredictably due to weather and other factors, and other changes occurring simultaneously could also impact population health and air pollution. When regulations are introduced to limit industrial pollution, it must be borne in mind that several other changes may occur in the background: an increase in traffic and an upgrade of residential heating systems, for example, or an economic recession leading to a reduction in pollution [30].

Exposure to pollution can vary, even within the same city, following a spatial distribution with significant gradients [31].

In a study conducted by Jerrett et al. in 2005 in the Los Angeles area, the chronic health effects found by considering intra-urban gradients of PM2.5 exposure were consistently higher than previously reported by studies using contrasts between average concentration values in different urban areas [32]. Finally, the study showed a consistent specificity of these health effects, with a stronger association between air pollution and mortality from myocardial infarction than from general cardiovascular or neurological causes or from all other causes [33]. 

In particular, the combined use of monitoring measures on an appropriate spatial scale and modeling techniques using continuous measurements of ultrafine particulate matter is important for the development of new technologies leading to early warning and control of any acute pollution phenomena linked to adverse weather phenomena [34].

The smallest particles penetrate the respiratory system at various depths and can take a long time to be removed, making them the most dangerous. They can reach the alveoli of the lungs, possibly leading to absorption into the bloodstream and intoxication, or aggravate chronic respiratory diseases such as asthma, bronchitis, and emphysema [35].

Evidence of health effects induced by exposure to particulate matter is provided by two types of studies:-**Epidemiological studies**, which aim to identify possible associations between concentrations of particles in the atmosphere and effects such as mortality, acute effects, effects on sensitive individuals (children and the elderly), etc., may be cross-sectional, retrospective, or prospective [36]. Cross-sectional surveys are of limited scientific value as they compare health indicators and exposure to pollutants in two different populations living in different areas. The prospective ones are scientifically very robust but very difficult and complex to implement. Retrospective studies use the time-series method, evaluating mortality and daily pollution with a lag interval of usually 24 h. It is interesting to note that, despite their relative simplicity, time-series studies provide data very similar to prospective studies, although they underestimate the real impact of PM exposure on mortality by 10–20% [37];-**Toxicological studies**, mostly of an experimental nature, aim to identify and understand the biological mechanisms by which exposure to particles can cause harmful effects in humans.

In recent years, epidemiological studies have been carried out to investigate the effects of exposure to fine and ultrafine aerosols in the short and long term [38].

The data show that the risk factor (relative risk between exposed and unexposed samples) is appreciably greater in populations exposed to fine and especially nanoparticulate particles [39]. In the assessment of health effects, the application of statistical methods makes it possible to eliminate or at least mitigate confounding effects (smoking habits, pre-existing lung diseases not attributable to dust, the role of other pollutants, etc. [40]). Due to their ability to penetrate the pulmonary alveoli and their enormous surface-to-mass ratio, it is possible that these particles inhibit the action of alveolar macrophages, which have the role of cleaning the lung tissue of inhaled foreign substances and are therefore called “sweeping cells”. It is also possible that macrophages, due to the presence of an overload of ultrafine particulate matter, destroy the material and increase the production of cytotoxic mediators (oxygenated reactive species, enzymes, and toxins), leading to increased susceptibility to infection, inflammation, and lung damage [41]. 

Of particular interest is the ability of pulmonary alveolar macrophages to release thromboxane with a prothrombotic action when their membrane is disrupted by PM [42]. This mechanism underlies the increase in plasma viscosity and measurable thrombophilia in sensitive individuals when exposed to PM. Interestingly, this increase is not generalized to the whole population but only affects less than 10% of subjects who are therefore defined as “susceptible” to the adverse effects of pollution. The flattening (plateau) of the relationship between PM and mortality at high doses of PM is related to the phenomenon of depletion of susceptible subjects [37]. Thus, the mortality/PM correlation is linear at low and intermediate doses but tends to plateau for values above 100 µg/m^3^. This situation explains the surprisingly attenuated impact of PM exposure at the very high PM values found, for example, in Asia. A major factor identifying susceptible individuals is the presence of atherosclerotic plaques [43]. 

In fact, the increase in thrombophilia induced by PM has no adverse consequences on the cardiovascular system if the arterial endothelium is structurally and functionally intact, thus being able to inhibit the activation of Hageman coagulation factors. In the case of subjects susceptible to the adverse effects of PM, on the other hand, the endothelium is absent due to the presence of diffuse atheromas lesions; the absence of endothelial inhibition, therefore, allows the increase in blood viscosity to manifest itself in the phenomena of arterial thrombosis that materialize, after an interval of 24 h following exposure, the increased incidence of obstructive cardiovascular events (e.g., myocardial infarction) in the susceptible population exposed to PM [44].

Since it is obviously not possible to assess the presence of atheromatosis in the general population, the age of the population is usually used as a proxy for this situation. Age is, in fact, an important risk factor for atherosclerotic disease, which develops as a chronic-degenerative disease. Therefore, elderly populations are much more at risk for the effects of exposure to pollutants on health than demographically younger populations [45].

Further tissue damage results from ultrafine particulate matter, which can adsorb strong acids, such as sulfuric acid (derived from sulfur oxides in the atmosphere), damaging alveolar tissues. In fact, strongly reducing pollutants, such as sulfur dioxide, are very reactive and directly damage the mucous membranes of the upper respiratory tract, thus constituting a risk factor for the onset of conjunctivitis, pharyngitis, and bronchitis [46].

However, the presence of small particulate matter, which acts as a carrier for these pollutants, can widen their range, allowing them to reach deep into the respiratory tract, with potentially more serious clinical consequences than those normally induced by these pollutants. The presence of metals (iron, manganese, vanadium, or nickel derived from combustion or platinum, palladium, and rhodium present in catalytic converters in motor vehicles) on the surface of the particles increases tissue irritation and allows their transfer to cells [47]. In addition, their catalytic effect promotes the formation of tissue-damaging oxidants in situ. 

The body response depends on the composition of the atmospheric dust and how the compounds are transported into the lung fluids. A large proportion of ultrafine organic particulate matter is soluble in water. This has important implications in determining the extent of exposure and toxicological mechanisms since water constitutes 90% of the fluids circulating in the lungs. While insoluble materials are internalized by phagocytic cells, the soluble fraction can alter lung fluid composition and be absorbed by tissues or enter directly into the lymphatic and circulatory systems [48]. Organic carbon is shown to have a specific activity in the fluids present within the alveoli. Solubility in lung fluids, reactivity, ability to form free radicals, the concentration of endotoxins, and hygroscopicity of particulate matter are properties of organic dust that are important for understanding the consequences induced by this pollutant on human health [49].

Observations made on different aerosol distributions provide the following information: aerosols of recent accumulation are retained after being released into the environment, their size distribution (0.1–0.3 mm) does not change much in urban-scale transport and is characterized by the presence of elementary tracers that differentiate the sources and accumulation is strongly influenced by relative humidity. The aerosol contains heavy metals that have long residence times and are deposited in the human respiratory tract [50].

Research in the United States and Europe shows that cumulative exposure to air pollution reduces lung development in children, accelerates the degeneration and aging of lung function in adults, increases the occurrence of chronic respiratory symptoms, and may also lead to a higher incidence of lung cancer in adults [51]. 

All these effects taken together lead to a reduction in life expectancy. The effects of air pollutants on human health are associated with an increase in respiratory diseases, a decrease in lung function indices, and the risk of cancer and leukemia, mainly due to PM2.5. PM10 would remain responsible for upper respiratory tract symptoms such as chronic bronchitis and bronchial asthma [52].

Acute episodes of pollution cause mild clinical effects in the healthy adult population, with a small reduction in lung performance, which the individual may not even notice, but which are of great epidemiological importance and great impact on public health, causing an increase in the number of population classes with reduced respiratory function [53]. 

The studies carried out show that it is not possible to define a particulate concentration threshold below which there is no effect on health. For this reason, it is not sufficient to assess the average exposure to PM over a long period of time (years, months, days), but it is necessary to carry out continuous accurate monitoring that shows the real risk induced by exposure to airborne PM [54]. 

Indeed, the considerable base of health and environmental data on airborne particles in working and living environments has provided indications of the characteristics that may influence toxicity and the dose-response relationship [55]. 

All these pathogenic mechanisms are triggered in the lung by point exposures to PM and not by the accumulation of PM in the respiratory system that occurs over a long period of time [56].

### 3.1. Respiratory Diseases

The respiratory system is the first route of entry into the body for air pollution, and the PM0.1 fraction also passes through the gastrointestinal tract by ingestion and can stimulate immune responses through the colon [57].

PM is scavenged in the lung by pulmonary alveolar macrophages that go up the tracheobronchial by the epithelial lining fluid clearance, finally arriving in the laryngo-pharynx and then in the esophagus [58,59,60]. This situation represents a detoxification mechanism due to the high detoxifying activities characterizing the liver at variance with the lung.

Fine PM induce oxidative damage in the skin, as demonstrated by measuring oxidized nucleotides in children [61].

A mucociliary respiratory system is an important tool for clearing inhaled particles. Although ultrafine particles can be trapped in the mucous layer, their role in clearance is far less than that for larger PM affecting the airways. Many pulmonary conditions impair mucociliary function. Mucociliary dysfunction is typical of smokers and individuals with respiratory tract infections: this explains the greater vulnerability of these individuals to air pollution [62]. Indeed, some different components of cigarette smoke, such as aromatic alcohols and phenols, are specific inhibitors of ciliary movements of the simple ciliated bathyprismatic epithelium lining the bronchi [63]. 

Another frequent situation of mucociliary clearance blockage that makes individuals particularly susceptible to the adverse effects induced by exposure to fine particulate matter is influenza syndrome: infection with Orthomyvoirus induces necrosis of the bathyprismatic epithelium, which is usually confined to the superficial epithelial monolayer unless it is a particularly lethal viral variant such as avian or porcine. However, these ciliated bathyprismatic cells are highly differentiated and specialized. Their reconstruction from the basal cells of the respiratory epithelium takes as long as two weeks [64]. 

During this period of convalescence, after the acute symptomatic phase of influenza, the patient is therefore particularly vulnerable to the effects of airborne pollution, particularly PM and the reducing pollutants carried by ultrafine PM. The flu syndrome is characterized by a strong seasonality with peaks in January–February, precisely when levels of PM and airborne reducing pollutants are at their highest. This phenomenon contributes significantly to fluctuations in general mortality on an annual basis, which peaks just a fortnight after the peak of the influenza epidemic. It is therefore of primary importance to improve the method of monitoring fine and ultrafine PM, especially in the first quarter of the year, by switching from average to point assessment. This progress would be of great importance for the management of public health issues related to PM exposure [65].

Particulate air pollution causes increased mortality in people with obstructive pulmonary disease (COPD), but the role of PM0.1 is not well understood. A Scottish study did not find PM0.1 to be more harmful than PM10, but other studies have reported that indoor biological PM0.1 in the form of bacterial extracellular vesicles causes inflammation and emphysema [66].

Toxicological evidence seems to support three hypotheses for the etiopathological mechanism underlying the effects on the respiratory system, namely the pulmonary inflammatory process; increased bronchial reactivity and exacerbation of asthma; and impairment of pulmonary defense mechanisms with increased susceptibility to infection.

Convincing evidence has emerged from experimental animal studies that, for the same material, ultrafine dust has a greater capacity to induce pulmonary toxicity than larger dust, also with reference to the greater capacity of particle deposition in the pulmonary alveoli and penetration into the interstitium. However, the key factor involved in this increased toxicity appears to be the increase in particle surface area, which coincides with the reduction in particle size [67].

Attention was drawn to the following factors involved in determining the response of the respiratory tract to exposure to ultrafine dust:-The surface activity of the particles and the ability of the particle surface to generate free radicals;-The particle aggregation/disaggregation capacities and the concentration of particles on the alveolar surface once the particles have entered the respiratory tree;-The ability to act as a carrier for different chemicals.

These factors can, in turn, be related to the size of the surface area but also to the intrinsic properties of the specific particles, especially if they contain chemicals of known high toxicity (e.g., TiO_2_) [68,69].

A clear association between increased mortality and episodes of heavy pollution caused by thermal inversion has been found in past years (London 1952 and 1956, New York 1963, etc.), and epidemiological studies now show a correlation between respiratory diseases and air pollution [70]. While the introduction of standards, control, and more refined technologies has reduced emissions of suspended dust and associated emissions of SO_2_ and NOx, the number of illnesses and mortalities associated with ultrafine dust pollution has not decreased proportionally. Moreover, studies carried out on historical series in cities with different climates and populations show that there is no threshold below which no effects are observed and that these effects are, in any case, correlated in a short time with changes in the concentration of pollutants in the atmosphere. The chemical and physical properties of particulate matter and of the smallest measurable fraction of particulate matter (nanoparticles), considered the most likely cause of health effects, are the subject of the most recent studies; from an epidemiological point of view, the correlation between fine particulate matter is statistically significant, but the correlation with ultrafine particulate matter is even more significant [71]. In the 2000 study, the WHO replicated a 1999 study in Italy, observing eight Italian cities with a population of over 400,000 inhabitants (Turin, Genoa, Milan, Bologna, Florence, Rome, Naples, and Palermo), corresponding to 15% of the Italian population. The results regarding risk factors are like those of the tripartite study. In the eight sample cities, the annual average concentration in 1998–1999 was above 40 mg/m^3^: from a minimum of 44.4 (Palermo) to a maximum of 53.8 (Turin). The study’s projections also predict a reduction of 5500 deaths per year if PM10 were to be reduced to 20 mg/m^3^ [72]. 

However, there may be host conditions, for example, in the case of sensitized organisms or organisms in poor health, particularly respiratory or cardiovascular disorders, that predispose them to suffer certain health effects not seen in healthy individuals. A recent study found that B vitamins may play a key role in reducing the impact of air pollution on the epigenome. Researchers gave 10 adults aged between 18 and 60 years a placebo or B vitamin (2.5 mg folic acid, 50 mg vitamin B6, and 1 mg vitamin B12) every day. To participate in the tests, volunteers had to be non-smokers, healthy, and not take any medications or vitamin supplements [73]. Analyses carried out on volunteers before and after taking placebo or vitamin B showed that vitamin B significantly increased median plasma concentrations of folic acid, vitamin B6, and vitamin B12. For those taking a placebo for 4 weeks, median plasma concentrations were like before. The particulate matter was collected from a busy street in downtown Toronto, through which more than 1000 vehicles per hour pass and inhaled by volunteers with an “oxygen-like” face mask. Several recent studies have found a relationship between exposure to environmental pollution and DNA methylation associated with respiratory function, genes related to cellular respiration, the cell cycle, and early development. Other studies have found shorter telomeres in children most exposed to air pollution, suggesting that telomere length may be a potential biomarker of exposure [74] (Figure 1).

### 3.2. COVID-19 and Atmospheric Particles

Several studies have confirmed that particulate matter plays a major role in the spread of COVID-19 infections. An Italian observational study demonstrated a significant positive relationship between COVID-19 incidence rates and levels not only of PM2.5 but also of NO_2_ (nitrogen dioxide) in Italy, both considering the period 2016–2020 and the months of the epidemic, correlated with two additional factors: the old age index and population density [75]. According to the analysis, exposure increases the COVID incidence rate by 2.79 sick people per 10,000 people if the concentration of PM2.5 increases by one microgram per cubic meter of air and by 1.24 sick people per 10,000 people if the concentration of NO_2_ increases by one microgram per cubic meter of air.

In fact, when our bodies are exposed to PM2.5, they develop a protein called ACE2 to defend themselves against this dust, but this protein increases susceptibility to SARS-CoV-2 infection by being the specific cellular receptor that allows binding to the virus spike protein, and thus, its entry into the cell [76].

An American study analyzed data from 3089 U.S. counties and concluded that an increase in just one microgram of PM2.5 per cubic meter of air is enough to lead to an 11% increase in mortality from coronavirus. The study also specifies that it is not the starting level of pollution in each area that is important, but rather the increase in fine particles. This means that even unpolluted areas with even a small increase in PM2.5 are at risk of increased mortality from COVID [77].

However, more recent studies have not confirmed the presence of SARS-CoV-2 in PM collected in some areas of the northeastern Po Valley [78]. However, this study used molecular probes for rather unstable regions of the virus, which are suitable for disease diagnosis on fresh oropharyngeal samples taken from subjects but not suitable for environmental monitoring [79]. This virus is, in fact, quite labile in environmental matrices and subject to degradation by bacterial and environmental RNases. What persists in the environment for a long time is often not the whole infecting virion but only fragments from the degradation of its nucleic acid. However, the rate of environmental degradation of SARS-CoV-2 depends on the environmental context. This virus is very sensitive to oxidizing agents and ultraviolet radiation. Oxidizing agents can rapidly neutralize the virus’ spike protein, which is characterized by a very high electrophilicity. This characteristic makes the virus particularly contagious, given the nucleophilicity of its cellular ACE2 receptor, but also particularly sensitive to oxidation typically induced by oxidants such as ozone [80] or oxidant species induced by UV exposure. The susceptibility of SARS-CoV-2 to oxidants contributes decisively to the strong seasonal fluctuations of the COVID-19 epidemic, with the lowest number of infections in the summer months characterized by strong oxidant pollution (ozone, nitrogen oxides) compared to the winter months characterized by strong reducing pollution (SO_2_, PM). In fact, the presence of reducing pollutants favors the maintenance of electrophilicity and, therefore, the reactivity of the virus spike protein, maximizing its contagiousness.

However, the relationship between COVID-19 and environmental pollution is complex and bivalent. In recent months, while the coronavirus emergency is dramatically clogging up intensive care units and causing thousands of deaths worldwide, China has seen a significant drop in nitrogen dioxide pollution and a marked improvement in air quality. Satellite images released by NASA leave no doubt and show a huge decline in pollution levels over China, due, at least in part, to the economic slowdown caused by the coronavirus because of restrictions imposed by the Beijing government on transport, commercial activities, and production levels in Chinese factories to contain the spread of the virus. NASA scientists say the reduction in rates of nitrogen dioxide in the air, a harmful gas emitted by motor vehicles and industrial facilities, was initially evident near the source of the outbreak, in the city of Wuhan, but then spread across the country at least comparing the first two months of 2019 with the same period this year. A drop in nitrogen dioxide levels had also been recorded in China and worldwide during the economic recession in 2008, but the reduction had been more gradual [81]. 

### 3.3. Cardiovascular Diseases

Many studies have shown an association between chronic exposure to PM0.1 and heart disease. A prospective study of 33,831 Dutch residents found that long-term exposure to PM0.1 was associated with an increased risk of cardiovascular diseases such as myocardial infarction and heart failure. In adults living in Toronto from 1996 to 2012, increased exposure to PM0.1 was associated with an increased incidence of heart failure and acute myocardial infarction. Particle monitoring, averaged over the year, found increased exposures associated with stroke, ischemic heart disease, and hypertension. Other studies have also found an increase in ischemic and thrombotic stroke with PM 0.1 exposure and increased blood pressure, and worse microvascular function with PM0.1 but not for PM2.5 and PM10 exposures [82]. It was also highlighted that exposure to metals contained in PM2.5 is associated with acute changes in ventricular repolarization as indicated by QT interval characteristics [83]. The QT interval (i.e., the electric heart repolarization interval as detected by electrocardiography) is related to heart rate in an inverse exponential relationship so that with the increasing rate, the QT interval shortens. Particle size is correlated with cardiovascular mortality: it was seen that the correlation becomes stronger as the particle size decreases (PM < 0.50 μm). Particle numbers are associated with cardiovascular disease with disease-related emergency room visits with a delay of 4–10 days; this is mainly due to particles of 10–50 nm. The strongest correlation of immediate effect (within 2 days) was found with particles of 30–100 nm, despite having a small mass concentration. The immediate effect related to mass concentration was seen with the 1–5 μm particles, which had a similar effect with PM0.1.

In general, the available information suggests a different systemic diffusion of ultrafine dust following inhalation exposure. The ability to pass through the alveolar epithelium is said to be size dependent, with a greater ability to absorb nanoparticles [84]. The systemic elimination mechanisms would also be size dependent, with greater difficulty in eliminating nanoparticles: in particular, alveolar macrophages are less efficient in eliminating them. Another hypothesis is that the nanoparticles manage to pass the capillary barrier and leave the circulation, resulting in systemic diffusion (translocation) due to their small size. Their distribution and localization within different organs probably also differ from that of larger particles, although the consequences in terms of toxicity expression are not obvious [85]. 

There are indications in the literature that the acute effects related to the number of ultrafine particles on respiratory health are more consistent than those related to the mass concentration of fine dust; a reduction in respiratory function and an increase in symptoms and the use of respiratory drugs have been highlighted as independent effects of ultrafine dusts, net of the effects related to fine dust; the acute effects of ultrafine dust on respiratory health are more consistent in adult asthmatics than in pediatric asthmatics [86]. Most studies agree that the most probable mechanism is related to the greater ability of these particles to induce an inflammatory response through chemical/reactive surface properties (the latter, as already mentioned, is considerably more extensive in ultrafine dust); this would be mediated by cytokines and triggered by oxidative damage to the respiratory epithelium and/or by impairment of the cellular signaling system that regulates cytokine synthesis. Both acute and chronic exposure is responsible for the pathogenesis of the damage. These effects would appear to be induced directly by the diffusion of particles in the bloodstream and/or indirectly through the action of neurochemical mediators; several pathogenetic mechanisms have been suggested, including increased blood viscosity, increased sympathetic “tone” and increased concentrations of peptides (e.g., endothelin) capable of inducing vasoconstriction, alteration of the heart rhythm, initiation of the thrombotic process, etc. These phenomena would be manifested in the presence of free oxidative radicals transported and generated by ultrafine particles passed into the circulatory stream or translocated into the organs, with a consequent inflammatory response, with a mechanism like that observed in the respiratory tree. The effects on health would be particularly harmful in people with compromised cardiovascular systems due to ischemic heart disease, cardiac rhythm disorders, and chronic obstructive pulmonary disease. In Italy, a study conducted between 2011 and 2015 examined 3338 children, showing that respiratory tract infections are more frequent in subjects living near heavily trafficked roads. According to World Health Organization (WHO) experts, high levels of PM2.5 alone cause up to 7 million premature deaths a year worldwide. In China, the number of fatalities is over one million, but some estimates even double the figure and put the number of deaths from environmental pollution at two million. 

Toxicological studies based on the concentration of atmospheric aerosols reveal interesting data: some guinea pig animals subjected to a concentrated atmosphere of fine particulate matter showed altered electrocardiograms and heart rhythm disorders that were fatal [87]. Toxicological studies carried out on a group of polycyclic aromatic hydrocarbons have confirmed that they cause skin tumors in mice and mammary cancers in rats. In the urban atmosphere, this class of compounds, which are adsorbed on the particle phase and largely on the fine and ultrafine particle fraction, is present at sufficiently high levels (0.01–1.0 mg/g aerosol) [88].

However, the lack of in-depth knowledge of the biological mechanisms of action on health hinders the development of effective control strategies for air pollution, and traditional toxicological studies using laboratory-generated aerosols with known particulate matter components are not yet able to determine in detail the mechanisms or causal agents responsible for the observed effects. Therefore, monitoring systems are needed to continuously determine the presence of airborne particulate matter (qualitative and quantitative) with respect to the prevention and control of this human health pollutant.

### 3.4. Neurological Damage and Nanoparticles

Pollutants deposited in the respiratory tract can reach the central nervous system via olfactory nerves, causing neurological damage with inflammatory mechanisms and oxidative stress, especially in children in whom the brain is not yet fully developed and is, therefore, more defenseless. According to a study carried out in Barcelona, primary school pupils with the highest levels of traffic pollution were found to have greater cognitive difficulties and neuro-behavioral disorders, particularly autism. Babies exposed during pregnancy to high levels of pollution may have slower intellectual development and lower IQ. A study conducted in northern Sweden found that pollution is a clear risk factor for neuronal inflammation in children: polluting particles cross barriers in the brain’s bloodstream and cause psychiatric disorders. Children living in areas of the city with more pollution were found to be prescribed more drugs for mental disorders, such as sedatives, sleeping pills, and antipsychotics [89]. A cross-sectional study in Milan shows that increasing levels of PM10 move the manic episode toward the depressive pole of the bipolar disorder spectrum and augment the probability of hospitalization for a manic episode with mixed components [90].

The neurological damage caused by pollution mainly affects the most fragile individuals. In addition to children and pregnant women, the elderly must also be included. Exposure to pollution also affects the nervous system of the elderly and promotes the development of dementia. In 2015, exposure to particulate pollution alone caused 4.2 million deaths worldwide, accounting for 7.6% of global mortality and ranking fifth overall as a major risk factor for mortality. Adding deaths caused by PM to those caused by excess ozone in the air and those caused by pollution from building heating systems, the total deaths reach 6.4 million in one year [91]. 

There is a discrepancy regarding the concentrations of pollutants not to be exceeded by law. The values indicated by Europe, and therefore by Italy, are generally higher than those indicated by the World Health Organization: for PM2.5, the annual limit is two and a half times higher than the recommended value and, despite the large number of deaths caused by this pollutant in Europe, not even the most recent legislative revisions have lowered it (Figure 2). 

## 4. Risk Assessment

As the correlation between ultrafine dust (expressed as several particles per cm^3^) and fine dust (expressed as mass concentration μg/m^3^) is rather weak, the statistically independent effect observed in multiple regression models is interesting. There is, therefore, epidemiological as well as toxicological evidence of biologically similar and independent responses to exposure to fine and ultrafine dust. However, given that both fine and ultrafine dust often originate from common sources and given the different dynamics of dust formation and accumulation and the different temporal relationships (lag-effects) observed between exposure and health effects, it is generally difficult to draw consistent causal inferences about the independent effects of the different types of dust. Given the observed variations in exposure over time to the different categories of particles (ultrafine vs. fine), it would theoretically be possible, in studies using continuous sampling and monitoring of time values of an appropriate size, to distinguish the independent effects of each component. Further studies could thus improve the understanding of the relationship between exposure to ultrafine dust and observed health outcomes. However, all available studies show that the main determinants of the effect of ultrafine particles are number and surface area and not weight. This means that the traditional use of gravimetric measurements of particulate matter is inappropriate for assessing the biological effects of ultrafine dust. The dynamics of ultrafine dust raise further doubts about the adequacy of individual monitoring stations (for measuring dust levels) inadequately representing levels for the entire geographical area in which the population resides. In particular, as the distance between the sources varies, especially for mobile sources such as vehicle traffic and individuals (in their homes or workplaces), exposure to ultrafine dust can change significantly, with reference to the sensitivity of these particles to different atmospheric stability conditions (atmospheric pressure, presence of winds, etc.) and their tendency to condense and aggregate, raising uncertainty about the representativeness of individual monitoring stations for exposure assessment. This assessment plays a key role in the *risk assessment* process: to correctly characterize and quantify the risk and health impact of a pollutant, it is necessary to obtain representative measurements and estimates of the actual concentration of the pollutant, with a detailed description of the entire distribution of pollutant concentration values over time and space. In addition, contrary to the literature on fine dust, there is not yet sufficient evidence of the correlation between ultrafine dust values measured by monitoring stations and those measured with personal exposure meters on population samples. Most of the studies on ultrafine dust indicate, as mentioned above, that vehicle traffic is the main source. However, vehicle exhausts, as well as those from combustion plants and many other emission sources, release a complex mixture of hundreds of components, both in the particulate and gaseous phases. It is, therefore, necessary to adequately characterize and quantify the composition of these mixtures to differentiate the effects of ultrafine dust from the effects of other components of ultrafine dust. 

A remarkable problem is represented by the identification of PM sources in complex situations characterized by multiple pollution sources. As an example, this problem has been faced by a recent paper in the Alcanti region in Spain, where cement and ceramic plants are located in the proximity of a highway [92].

## 5. Conclusions

To produce appropriate guidelines for the health risks assessment from nanoparticles, precise and systematic knowledge of the different levels of environmental and human exposure to ultrafine dust, including past exposure, needs to be completed through the development of new measurement techniques for *routine* use. It is important to learn more about the mechanisms and kinetics of nanoparticle release from each of the many emission sources involved. It is necessary to study and implement models that allow predictions about the release of nanoparticles from these sources, based on the above information, knowledge of the substances used, and how they are used. 

More generally, further studies are needed to identify the rules governing the toxicology and ecotoxicology of nanoparticles. In the results of these studies, it does not seem to be possible to assume, for the purposes of risk assessment, that a nanoparticle with a defined chemical composition has effects on biological systems that are comparable to those of the same chemical in other physical forms. One of the main questions to be answered by research is whether there are definite ways of exposing humans and the environment to adequately characterized nanoparticles. The exposure dose will need to be defined in terms of particle number (and possibly total surface area) rather than in conventional terms (mass concentration μg/m^3^), considering that a change in the size/shape and other physicochemical properties of a nanoparticle could result in changes in expected adverse effects. Conventional toxicity studies may require some modifications when carried out on nanoparticles to simulate realistic exposure scenarios and to define endpoints that are directly associated with the nanoparticles to be assessed.

In conclusion, the scientific data reported in the present review indicate that there is solid scientific evidence on the low predictivity of risk to health status as inferred from the average assessment of PM over an often very long period (daily, annual, and monthly averages). In fact, the averages interpolate the very high PM levels reached in some specific periods with the very low levels reached in other specific periods. For example, the daily average interpolates the very high values reached during the day with the very low values at night. Similarly, the annual average interpolates the high PM values of the winter months with the lower values of the summer months. The assessment of average PM, therefore, significantly underestimates the health risk of reaching high PM values even for limited periods of time. It, therefore, seems useful and necessary to develop new models and new technologies to monitor PM no longer on an inferential average basis but on an ad hoc basis. The existing legislative regulation of air pollution is mainly focused on establishing threshold levels as related to average values evaluated on a long time basis (days, hours). The development of new analytical methods allowing timely PM analysis represents an important perspective for PM control.

## Figures and Tables

**Figure 1 ijerph-19-10587-f001:**
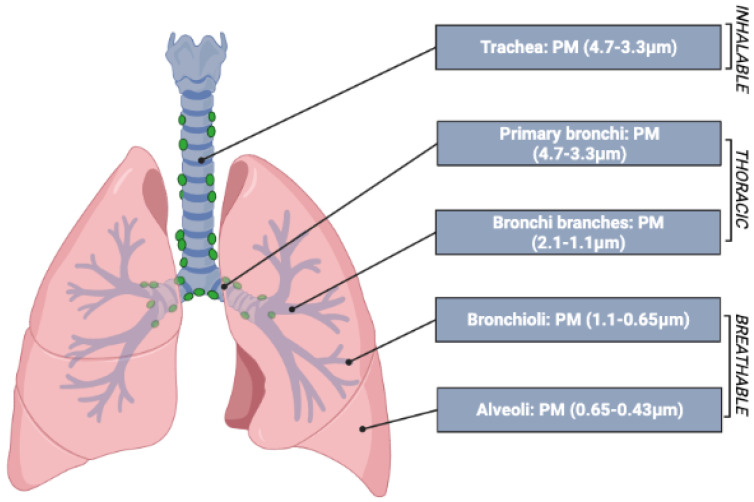
PM penetration level in the respiratory system.

**Figure 2 ijerph-19-10587-f002:**
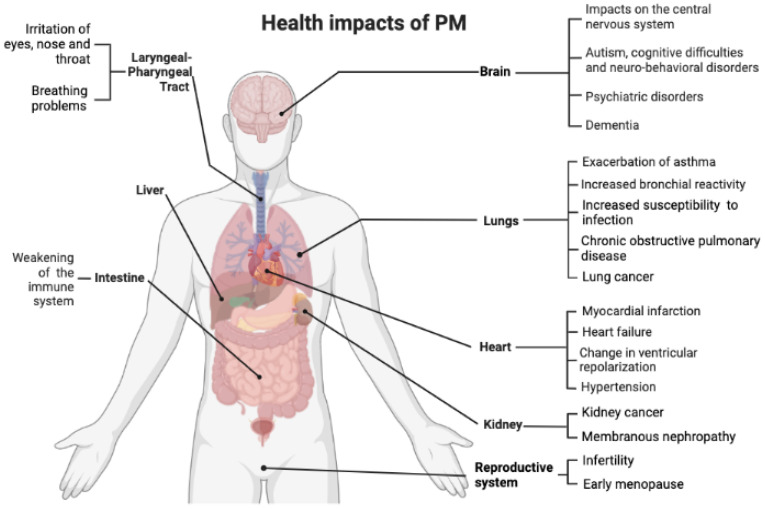
Health impacts of air pollution.

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
