# Peer review of "Importance of Punctual Monitoring to Evaluate the Health Effects of Airborne Particulate Matter"

_ijerph, 2022, doi:10.3390/ijerph191710587_

Round 1

Reviewer 1 Report

Overview and General Impression
The presented manuscript (MS) is dedicated, as stated by the authors, on the investigation of the causes of PM pollution and classification the different types of dust based on their size. Indeed, the work could be described, most concisely, as review of the contemporary epidemiological and more specifically human health-related aspect of the problem of the ambient air pollution with (non-mineral) aerosol. Due to the profound importance of the question, the subject of the MS is out of any doubt. It is also relatively well motivated by the authors, especially in the local context. It fits also well in the scope of ‘Int. J. Environ. Res. Public Health’. The work is with good concept and acceptable structure and length. Subsequently, it has the potential to generate interest. As main strength of the study, alongside the practical importance, I could outline the consideration of the Covid-19-related issues.
The MS suffers, however, from some weaknesses which could be addressed before the MS becomes publishable.

Major remarks
My main concern stems from the absence of clear formulation of the aims and goals of the study (which should be stated in the Introduction) and emphasized outcomes (in the Conclusion). In fact, alongside the in-depth and skillful synopsis of the already documented knowledge, what are the key new messages, respectively what is the novelty, of the work? The conclusion contains (as seems well-grounded) implicit criticism to the existing legislative regulation of the air pollution, but what are the proposals, at least tentative and preliminary, for concrete changes? They should be at least realistic, i. e. conformable with the existing measurement networks/facilities, etc.
Another my concern is caused by the lack and/or clear definition of the main term of the work, namely particulate matter (PM). In fact, PM10 appears on row 34 and there (as elsewhere) there is no any clarification what the number after ’PM’ means. Second, which is much more important, the  explicit definition of the fractions PM10 and PM2.5 according Article 2 point 18 and 19 of the cited Directive 2008/50/EC (attached in this review) should be explained in order for right understanding of the rest of the MS. Thus it should be clarified (keeping in mind that the main reader’s target group are non-physisist), that any PM plum/population is not mono-modal, i.e. the characteristic size distribution is spread across orders. Thus, the ‘coarse mode’ PM10 still contains amount of small enough particles and should be not miss-interpreted as ‘health-save’. Further, the correct term is ‘aerodynamic diameter’ (row 109) instead of ‘diameter’ (hence the particles are fairly non-spherical).
These questions should be explicitly clarified in the revised version.

Specific remarks
r325: TiO or TiO2? Check!
r335: bronchi[… → blank is missing.
r352: exposure[… → blank is missing.
r397-399 mg/m3-→micro_sign g/m3
r464&468: Nasa -->NASA
r475&r477: PM 0.1 → PM0.1 (remove blank)
r 484: PM 10 → PM10 (remove blank)
r 485: What is QT?

Author Response

We wish to thank the Editor and the Reviewers for their thoughtful and comprehensive review of our manuscript. We have done our best to address every concern, and we believe that the revisions made have considerably strengthened the manuscript. Our responses to the Reviewers’comments and the related changes made to the manuscript are listed below.

We have also updated the text accordingly with the suggestions of the reviewer’s, thus further improving the coverage and completeness of our review. We are most grateful to the IJERPH for the opportunity to submit this revised manuscript.

Reviewer #1

My main concern stems from the absence of clear formulation of the aims and goals of the study (which should be stated in the Introduction) and emphasized outcomes (in the Conclusion). In fact, alongside the in-depth and skillful synopsis of the already documented knowledge, what are the key new messages, respectively what is the novelty, of the work?

R: We really appreciate the Reviewer’s suggestion, for this reason we expained the aims, the outcomes and the novelty of the work in the introduction and in the conclusions as suggested. 

The conclusion contains (as seems well-grounded) implicit criticism to the existing legislative regulation of the air pollution, but what are the proposals, at least tentative and preliminary, for concrete changes?

R: We thank the Reviewer for the valuable suggestion. We revised the text and we added in the conclusions as requested.

Another my concern is caused by the lack and/or clear definition of the main term of the work, namely particulate matter (PM). In fact, PM10 appears on row 34 and there (as elsewhere) there is no any clarification what the number after ’PM’ means. Second, which is much more important, the explicit definition of the fractions should be explained. Thus it should be clarified (keeping in mind that the main reader’s target group are non-physisist), that any PM plum/population is not mono-modal, i.e. the characteristic size distribution is spread across orders. Thus, the ‘coarse mode’ PM10 still contains amount of small enough particles and should be not miss-interpreted as ‘health-save’. Further, the correct term is ‘aerodynamic diameter’ (row 109) instead of ‘diameter’ (hence the particles are fairly non-spherical).
These questions should be explicitly clarified in the revised version.

R: We thank the Reviewer for the note. We have revised the text accordingly to reviewer’s suggestions.

Specific remarks
r325: TiO or TiO2? Check!
r335: bronchi[… → blank is missing.
r352: exposure[… → blank is missing.
r397-399 mg/m3-→micro_sign g/m3
r464&468: Nasa -->NASA
r475&r477: PM 0.1 → PM0.1 (remove blank)
r 484: PM 10 → PM10 (remove blank)
r 485: What is QT?

R: We thank the Reviewer for the remark and we are sorry for the mistakes. We corrected and update the text.  The QT interval (electric heart repolarization interval as detected by electrocardiography) is related to heart rate in an inverse exponential relationship, so that with increasing rate the QT interval shortens. This clarification has been added in the text.

Reviewer 2 Report

This paper reviews the heath effect of particular matters, which is an important topic for us to develop proper policy and strategy to control the PM level. The authors also underscore the importance to develop more proper short-term standards for PM level. However, I have detected few directly copy from other manuscripts by using iThenticate (e.g., L311-313 and L320-327 were directly copied from “The health effects of ultrafine particles”, L84-87 were directly copied form “A review on the human health impact of airborne particulate matter”, etc.). Besides plagiarism, this manuscript also did not put necessary references (e.g., L48-53, “direct emissions … fires.”, L108-109, “to date, … 10 μm”, L430-433, etc.). Furthermore, there are many references cited in the manuscript that were out of date, and the authors cited many review papers. Moreover, the author of this topic is very important, there are already lots of review paper focused on the similar topic (e.g., " Impact of fugitive emissions in ambient PM levels and composition: A case study in Southeast Spain”). Therefore, I suggest the editor reject this manuscript.

Author Response

This paper reviews the heath effect of particular matters, which is an important topic for us to develop proper policy and strategy to control the PM level. The authors also underscore the importance to develop more proper short-term standards for PM level. However, I have detected few directly copy from other manuscripts by using iThenticate (e.g., L311-313 and L320-327 were directly copied from “The health effects of ultrafine particles”, L84-87 were directly copied form “A review on the human health impact of airborne particulate matter”, etc.). Besides plagiarism, this manuscript also did not put necessary references (e.g., L48-53, “direct emissions … fires.”, L108-109, “to date, … 10 μm”, L430-433, etc.). Moreover, the author of this topic is very important, there are already lots of review paper focused on the similar topic (e.g., " Impact of fugitive emissions in ambient PM levels and composition: A case study in Southeast Spain”).

R: Sentences citing text as mentioned by the reviewer has been removed. We have updated references in the revised version as suggested.

Reviewer 3 Report

The article take into account a very important topic. The mechanisms regarding the different health effects and exposure were well described.

 Attention dedicated to the dimensions of particles was appreciated, as well as to the WHO guidelines on air quality. 

 I have no comments about the form that is well written.

 Revisions:

Line 33: Public Health: replace with: public health

Lines 34, 35, 36, 40, 41, 233, 397, 398, 582: μg/m3 replace with: μg/m3

Line 97:

In addition, when considering the WHO guideline values for population exposure to replace with:

In addition, when considering the WHO guideline values for population exposure to replace with

 Line 296: tion. [52]  replace with: tion [52].

 Line 313: biopsies. [56].  replace with: biopsies [56].

 Lines 319-320: system. [57].

In the Western diet, more than 1012 ultrafine particles are ingested daily by a

replace with: system [57]. In the Western diet, more than 1012 ultrafine particles are ingested daily by a

 Line 321: single person. [58]. replace with: single person [58].

 Line 374:                                                     - the ability to act as a carrier for different chemicals.

replace with:

- the ability to act as a carrier for different chemicals.

 Line 417: [73]. (Figure 1).  Replace with: [73] (Figure 1).  

 Line 424: NO2 (nitrogen dioxide) replace with: NO2 (nitrogen dioxide)

Line 444: monitoring. [77]. 444 replace with: monitoring [77].

Line 577: it. (Figure 2). replace with: it. (Figure 2).

References

correct alignment: 8-72; 74; 77; 79-80

Author Response

The article take into account a very important topic. The mechanisms regarding the different health effects and exposure were well described.

Attention dedicated to the dimensions of particles was appreciated, as well as to the WHO guidelines on air quality. 

 I have no comments about the form that is well written.

R: We wish to thank the Reviewer for the kind and positive comments. We have revised the manuscript, taking into account the Reviewer’s concerns below.

Line 33: Public Health: replace with: public health

Lines 34, 35, 36, 40, 41, 233, 397, 398, 582: μg/m3 replace with: μg/m3

Line 97:

In addition, when considering the WHO guideline values for population exposure to replace with:

In addition, when considering the WHO guideline values for population exposure to replace with

 Line 296: tion. [52]  replace with: tion [52].

 Line 313: biopsies. [56].  replace with: biopsies [56].

 Lines 319-320: system. [57].

In the Western diet, more than 1012 ultrafine particles are ingested daily by a

replace with: system [57]. In the Western diet, more than 1012 ultrafine particles are ingested daily by a

 Line 321: single person. [58]. replace with: single person [58].

 Line 374:                                                     - the ability to act as a carrier for different chemicals.

replace with:

- the ability to act as a carrier for different chemicals.

 Line 417: [73]. (Figure 1).  Replace with: [73] (Figure 1).  

 Line 424: NO2 (nitrogen dioxide) replace with: NO2 (nitrogen dioxide)

Line 444: monitoring. [77]. 444 replace with: monitoring [77].

Line 577: it. (Figure 2). replace with: it. (Figure 2).

References

correct alignment: 8-72; 74; 77; 79-80

R: We thank the Reviewer for the note. We revised the text as suggested. The revision are reported in the text in Yellow colour.

Round 2

Reviewer 2 Report

I do not see any revision based on my previous comments. Significant plagiarism has been detected. Also, there are too many similar studies (see my previous comments). 

Author Response

We appreciated the Reviewers’ comments, and we revised the manuscript accordingly. Please find enclosed the revised version of the manuscript and the point-by point reply to the Reviewer’s comments. For the sake of clarity, changes in the revised manuscript are highlighted in light blue.

According to Reviewer’s comment, we have added references as indicated in the old version in lines 48-53, now in the last version in lines 54-62.

We have changed lines 84-87, as suggested by the reviewer for plagiarism, now reported in lines 99-100.

In this last version, as suggested by the Reviewer we have added reference in line 120 and reference in line 452 (previous version in lines 108-109 an lines 430-433, respectively).

As suggested, we have changed lines 311-313 (old version) in lines 324-327 (new version),  lines 320-327 (old version) in lines 324-327 (new version), lines 323-327 (old version) in lines 345-346 (new version). Please see the attachment.

We hope that the new revised version of our manuscript will be now suitable for publication in IJERPH. Thank you for Editor’s and Reviewers’ support and suggestions helping us to improve our manuscript.
